Time series (ARIMA) as a tool to predict the temperature-humidity index in the dairy region of the northern desert of Mexico

Herrera-González José Luis 1
http://orcid.org/0000-0002-7670-9425 Rodríguez-Venegas Rafael 2
http://orcid.org/0000-0001-6134-0218 Legarreta-González Martín Alfredo 3 4
Robles-Trillo Pedro Antonio 5
De-Santiago-Miramontes Ángeles 5
Loya-González Darithsa 3
Rodríguez-Martínez Rafael 2 rafael.rdz.mtz@gmail.com
1 Programa de Doctorado en Ciencias Agropecuarias, Universidad Autónoma Agraria Antonio Narro , Torreón, Coahuila , Mexico
2 Departamento de Ciencias Médico Veterinarias, Universidad Autónoma Agraria Antonio Narro, Unidad Laguna , Torreón, Coahuila , Mexico
3 Universidad Tecnológica de la Tarahumara , Guachochi, Chihuahua , Mexico
4 University of Makeni, Sierra Leone , Makeni , Sierra Leone
5 Departamento de Producción Animal, Universidad Autónoma Agraria Antonio Narro, Unidad Laguna , Torreón, Coahuila , Mexico
Wang Xinfeng
Electronic publication date: 2024 Dec 18
Publication date: 2024
Volume: 12
Electronic Location ID: e18744
Received 2024 Sep 20; Accepted 2024 Dec 2
Copyright: © 2024 Herrera-González et al.
Copyright year: 2024
Copyright holder: Herrera-González et al.
License: This is an open access article distributed under the terms of the Creative Commons Attribution License, which permits unrestricted use, distribution, reproduction and adaptation in any medium and for any purpose provided that it is properly attributed. For attribution, the original author(s), title, publication source (PeerJ) and either DOI or URL of the article must be cited.
License URL: https://creativecommons.org/licenses/by/4.0/

Keywords: ARIMA, Temperature-humidity index, Heat stress, Dairy cows, Arid zone, Mexico, Calibrated models

Funding: Universidad Autónoma Agraria Antonio Narro 38111-425502002-2742 Consejo Nacional de Humanidades Ciencia y Tecnología (Conahcyt) 928570 This work was supported by Universidad Autónoma Agraria Antonio Narro, (Grant Number 38111-425502002-2742). Author Herrera-González, José Luis has received research support from Consejo Nacional de Humanidades Ciencia y Tecnología (Conahcyt), (Grant number 928570). The funders had no role in study design, data collection and analysis, decision to publish, or preparation of the manuscript.

==============================
The environment in which an animal is situated can have a profound impact on its health, welfare, and productivity. This phenomenon is particularly evident in the case of dairy cattle, then, in order to quantify the impact of ambient temperature (°C) and the relative humidity (%) on dairy cattle, the temperature-humidity index (THI) is employed as a metric. This indicator enables the practical estimation of the stress imposed on cattle by ambient temperature and humidity. A seasonal autoregressive integrated moving average (SARIMA) (4,1,0)(0,1,0)365 model was estimated using daily data from the maximum daily THI of 4 years (2016–2019) of the Comarca Lagunera, an arid region of central-northern Mexico. The resulting model indicated that the THI of any given day in the area can be estimated based on the THI values of the previous four days. Furthermore, the data demonstrate an annual increase in the number of days the THI indicates a risk of heat stress. It is essential to continue building predictive models to develop effective strategies to mitigate the adverse effects of heat stress in dairy cattle (and other species) in the region.

Introduction

The negative impact of heat stress (HS) on livestock productivity has been well documented in the literature. Among others, the studies by Armstrong (1994), Kadzere et al. (2002), Amundson et al. (2006), Salem & Bouraoui (2009), Gantner et al. (2011) and Hernández et al. (2011) have all highlighted the detrimental effects of HS on animal thermoregulation and feed intake, fertility, and milk production. High-yielding animals are particularly susceptible to HS due to their elevated thermogenesis, which is a consequence of their heightened metabolic activity (Bernabucci et al., 2014; St-Pierre, Cobanov & Schnitkey, 2003). As dairy cows are primarily selected for their milk production, they are more susceptible to caloric stress, which has been demonstrated to significantly impair their fertility (Sammad et al., 2020). Furthermore, it has been demonstrated that in dairy cows increase their milk production from 35 to 45 kg/d, the temperature threshold for HS can be lowered by 5 °C. This indicates that higher milking cows are susceptible to HS at lower temperatures (Armstrong, 1994). Consequently, the dairy industry incurs economic losses due to heat stress. In the United States, the financial impact of heat stress is estimated to range from 897 to 1.5 billion dollars annually (St-Pierre, Cobanov & Schnitkey, 2003).

A variety of bioclimatic indices have been utilized as a means of predicting the HS and its impact on dairy cattle. Of these indices, the temperature-humidity index (THI) is the most utilized and practical. Its origins can be traced back to studies conducted in the 1940s. Moreover, it has been a valuable indicator of heat stress in dairy cattle since the 1960s (Vasseur et al., 2012). Since that time, the THI has been employed to assess the productive and reproductive response as a function of climate differences (Hahn, Mader & Eigenberg, 2003; Ravagnolo, Misztal & Hoogenboom, 2000; Silva, Morais & Guilhermino, 2007; Tolkamp et al., 2010). The THI is a practical bioclimatic marker that reflects the sum of external forces acting on animals (temperature and humidity) and their impact on body temperature homeostasis (Silva, Morais & Guilhermino, 2007). The THI is calculated using a variety of formulas developed from research that measure dry bulb, wet bulb, dew point temperatures, and relative humidity of the air (Sejian et al., 2013).

As previously proposed by Houlahan et al. (2017), the objective of scientific inquiry is to gain an understanding of the natural world. The capacity to make predictions is the sole means of substantiating scientific comprehension, thereby establishing it as a foundational tenet of all scientific disciplines. In the modern era, prediction fulfills two vital functions. Firstly, prediction serves as a test of scientific understanding, thereby conferring authority and legitimacy upon it. Secondly, prediction may also function as a potential guide for decision-making (Sarewitz & Pielke, 1999).

In the Comarca Lagunera, situated in the northern arid region of Mexico, HS conditions are present throughout the year (305 d), exerting a detrimental impact on milk production, milk composition, cow comfort, and the ratio of milking cows to nonmilking cows. Furthermore, these conditions have the potential to impose an economic burden at the farm, regional, and societal levels (Rodriguez-Venegas et al., 2023). Mathematical models are employed by scientists to predict the potential consequences of natural phenomena, with the objective of developing strategies to mitigate their adverse effects. Among the aforementioned tools, the following may be identified: A time series can be defined as a collection of observations made sequentially over time, in a broad sense, and can be used to describe a variety of data sets. A time series can be defined as a specific type of stochastic process. The last decades have shown great progress in the technique and scope of the use of models in the biological sciences, however, in the area of farm animal welfare the variety, type, and complexity of the models used have not advanced at the same pace, despite the fact that they could have a great scope in this field of research (Collins & Part, 2013). The present study will focus on time series exhibiting behavior consistent with the laws of probability, as opposed to deterministic series. In the field of dairy production by cows, time series analysis has been applied in several areas, including the modeling of diseases such as estrus (de Mol et al., 1999), the quantification of the effect of temperature on mortality in dairy cows (Morignat et al., 2015), the increase in production due to dietary changes (Kerr, Cowan & Chaseling, 1991), the demand for dairy products (Heien & Wessells, 1988), and methane and CO2 production (Lee et al., 2017). It is evident that the applications of this methodology to explain and predict the phenomenology in agricultural issues are numerous and diverse. This enables the implementation of preventative measures in a timely manner, thereby preventing any adverse effects on the health, comfort, and productivity of cows. For this reason, this article examines the predictive capacity of the THI in relation to potential HS events in the Comarca Lagunera, employing the time series method.

Materials and Methods

The climate data from the Comarca Lagunera (102°22′, 104°47′ WL; 24°22′, 26°23′ NL, at 1,139 m) were the subject of this study. This arid region of northern Mexico accounts for 21% of the national dairy cow inventory and presents environmental conditions that present a significant challenge to Holstein cattle on dairy farms. These conditions include an average annual precipitation of 200 mm, extreme ambient temperatures that can range from −5 °C in winter to 41.5 °C in summer, and high solar radiation.

Ambient temperature (in degrees Celsius) and relative humidity (in percent) data were obtained to calculate daily THI using the DiGiTH™ application (DiGiTH Technologies, Mexico), from five representative geographical points (GPs), according to the process described in a previous study of Rodriguez-Venegas et al. (2022). The THI was calculated as (1.8T+32)−[(0.55−(0.0055×RH)((1.8T)−26)] (Council, 1971). GPs were as follows: GP1 is situated at 25.5° NL and 103.25° WL; GP2 is located at 25°61′ NL and 103°55′ WL; GP3 is situated at 25°90′ NL and 103°39′ WL; GP4 is situated at 25°51′ NL and 103°60′ WL; GP5 is located at 25°40′ NL and 103°31′ WL. The data set under consideration spanned the period from 2016 to 2019.

ARIMA model forecast

In reference to a specific time series, the predicted observation is calculated using the equation designated as Eq. (1) (Box et al., 2016).

(1) Yt=Y1+Y2+Y3+…+Yt

where Y is the observations in the time of t.

As mentioned by Patle et al. (2015), Eq. (1) into Eq. (2)

(2) Yt=c+ϕ1Yt−1+ϕ2Yt−2+…+ϕpYt−p+et.

In a study conducted by Gibrilla, Anornu & Adomako (2018), the two constants, c and phi1, were employed in an analogous manner to address random error in t. The aforementioned errors were accounted for through the utilization of the variable et.

(3) Yt=c+et−ϕ1et−1−ϕ2et−2−…−ϕqet−q.

Sen’s estimator

In general, the slope is employed for the assessment of linear patterns through the implementation of least squares estimation via linear regression. The slope estimation formula, as proposed by Sen (1968), is presented in the following equation:

(4) dk=xj−xij−i

where,

dk is an estimated slope.

for (1≤i<j≤n), where d is the slope, x denotes the variable, n is the number of data, and i, j are indices.

Sen’s slope is then calculated as the median from all slopes: bSen=median(dk).

Mann-kendall trend test

The Mann-Kendall test is a non-parametric method used for the analysis of trends in time series data, initially proposed by Kendall (1938). The alternative hypothesis postulates the existence of a monotonically increasing or decreasing trend.

(5) S=∑i=1n−1∑j=i+1nsign(xj−xi).

As presented by Anand et al. (2020), the Mann–Kendall statistic (S) is calculated from Eq. (6).

(6) sign(xj−xi)={+1if(xj−xi)>0,0if(xj−xi)=0,−1if(xj−xi)<0.

Data analysis

We used R (Version 4.4.1) (R Core Team, 2024), and the R-packages fable (Version 0.3.4) (O’Hara-Wild, Hyndman & Wang, 2024a), fabletools (Version 0.4.2) (O’Hara-Wild, Hyndman & Wang, 2024b), forecast (Version 8.23.0) (Hyndman & Khandakar, 2008), ggplot2 (Version 1423.5.1;27), (Hyndman & Khandakar, 2008), lubridate (Version 1.9.3;29) (Grolemund & Wickham, 2011), and trend (Version 1.1.6) (Pohlert, 2023) for all our analyses.

A variant of the Hyndman-Khandakar algorithm (Hyndman & Athanasopoulos, 2021) was used for model selection. This algorithm integrates unit root tests, Akaike information criterion minimization (AICc) and maximum likelihood estimation (MLE) to derive an ARIMA model. For the forecasting, a model calibration was used following the steps used by Legarreta-González et al. (2024a, 2024b) as follows:

Train/test

Spliting of the time series into two sets. Training.

Testing (12 months).

Visualization of the train/test split.

Modeling estimation, where the equation of the model was. m3∼date,training(splits).

Modeltime workflow

The objective was to accelerate the evaluation and selection of models in a systematic and efficient manner in a systematic and efficient manner, with a view to facilitating the process of identifying the most appropriate models for a given context. In light of the vast array of time series models currently available, it is feasible to undertake an analysis of these models and to forecast future outcomes by employing the model-time approach.

Workflow

1. Creation of a Modeltime table. The Modeltime table employs a system of identification numbers and the generation of generic descriptions to facilitate the organisation and monitoring of models.

2. Model calibration. The objective of model calibration is to quantify the extent of errors and to estimate confidence intervals. Model calibration was conducted on the test set.

3. Forecast using the testing set. 1) Calibration of the data permits visualization of the test predictions, which may be regarded as a forecast.

2) The subsequent step is to ascertain the accuracy of the testing process, thus facilitating comparison of the models.

4. Analyses of the results. The optimal model is selected based on an evaluation of the accuracy of the measures employed and the results of the forecasts, whereby the latter are assessed in terms of their predictive power.

5. Refitting. The final step was to perform a realignment in accordance with the established best practice prior to forecasting future outcomes.

Results

Two data sets were constructed using the most recent 12 months of data, thereby enabling the formation of a training and testing set utilizing the final 12 months of data as the testing set. The subsequent step was to estimate the models with the resulting estimates. Subsequently, the models were calibrated using the out-of-sample data, that is to say, the testing set. The resulting fitted values and residuals for the testing set, which permit a comprehensive examination of the data, were calculated for the purpose of calibration. The aforementioned data permit the generation of a visual representation of the testing predictions, or forecast. This calibration is consistent with works of Legarreta-González et al. (2024a, 2024b) performed in time series analyses.

A seasonal autoregressive integrated moving average (SARIMA) (4,1,0)(0,1,0)365 model was obtained with an R2 of 0.63. The model comprises an autoregressive component of order four, indicating that the value of the THI on a given day exhibits an autocorrelation of four previous days. Additionally, a differencing of one was necessary to achieve stationarity. Furthermore, a seasonal component was incorporated, where m corresponds to the 365 days of the year, also with a differencing of one.

The estimators of the model are as follows Eq. (7):

(7) τt=(−0.3149ϕt−1−0.2765ϕt−2−0.3036ϕt−3−0.1904ϕt−4+ϵtϵt∼NID(0,2.541)

The observed data (black line) and the model prediction (red line) in Fig. 1 exhibit a similar temporal behavior for THI, indicating that the model is performing well. This conclusion can be further substantiated in the subsequent section.

Figure 1 Time series for temperature humidity index (THI), with observed values in black, and predicted by the model in red.

In consideration of the statistical data that quantify the trajectory of the forecast, a trend is discernible, indicating that the model anticipates an escalation in THI. The slope is 0.01334708, with a 95% confidence interval of [0.01026692, 0.01629877]. The Mann-Kendall trend test statistic is S=1.910000×104. The variance is 5.425116×106, and the τ=2.875229×10−1. The z-score is 8.1999, the sample size is 365, p < 0.001.

Figure 2 illustrates that the residuals of the model exhibit a “white noise” behavior, as although some peaks are evident, they are not statistically significant. This is further corroborated by Fig. 3 showing the Autocorrelation Function (ACF) and Partial Correlation Function (PACF), which indicate that the model has been estimated correctly.

Figure 2 Residuals analysis of the model.

Figure 3 Autocorrelation function (ACF), and the partial autocorrelation function (PACF) from the model.

Discussion

This article presents, for the first time, the behavior of the THI using an ARIMA model, for which neither AR nor MA values had been previously estimated for this animal welfare indicator. Due to their great influence on key elements for the success of livestock farming, such as production, health, and animal welfare, environmental control systems are used to maintain a series of variables, such as temperature, humidity, and contaminant concentrations, at optimal levels. These systems are the most efficient tools to guarantee better production in livestock buildings (Besteiro et al., 2017) and tools such as ARIMA models are used for this purpose. In our case, this tool was used to evaluate the use of THI data and predict its future behavior. This would contribute to the opportune use of the mechanisms that allow avoiding or reducing the effects of heat stress on farm animals, mainly dairy cattle.

The design of THI prediction models consists of two stages: estimation and forecasting. Hyndman & Khandakar (2008) developed an algorithm for ARIMA model selection that uses the best AICc once the PAC/ACF requirements and stationarity are met. On the other hand, the geographic sites from which the data for the construction of the THI were obtained are representative of the region. Additionally, the climate data was as complete as possible, using a total of 1,404 THI data to design our ARIMA model.

The initial portion of the SARIMA model indicates that the THI of the present day is autocorrelated with the four preceding days. The coefficient for the previous day (t-1) to the THI to be estimated was −0.31, while the coefficients for t-2, t-3, and t-4 were −0.28, 0.30, and 0.19, respectively. A slope was observed, necessitating differentiation to achieve stationarity. No evidence of a moving average was observed, indicating that the THI of a given day can be predicted based on the model, with a confidence interval of 85%, by considering the THI values of four previous days. The second part of the model (0,1,0) allows for the detection of a seasonal pattern of 365 days, in which there is only a slope, but no AR or MA component.

With regard of the Sen’s slope was 0.01334708, Mann-Kendall trend test S=1.91×104 (z = 8.2, n = 365, p-value=2.407×10−16) indicating a monotonic trend which signifies the presence of a slope in SARIMA prediction. There is a correlation with previous reports indicating an increase in the number of days with THI levels that induce heat stress in dairy cattle. In this context, Reiczigel et al. (2009) reported an increase in the number of days per year experiencing thermal stress (THI ≥ 68), from 5 to 17, over the past 30 years. Similarly, Dunn et al. (2014) proposed that by the year 2,100, the number of days exceeding the THI threshold may increase from an annual average of 1–2 to over 20. Hempel et al. (2019) proposed that the impact of prospective increases in thermal stress risk will vary across locations. They posit that there will be a general trend towards an increase in the number and duration of thermal stress episodes. In their study of the Comarca Lagunera, Rodriguez-Venegas et al. (2022) observed an increase in the annual number of days with THI levels above the normal THI threshold (i.e., ≥68) over time. The observed increase in temperature has the potential to compromise the reproductive and productive soundness of Holstein cows in northern arid Mexico.

The application of ARIMA models has been demonstrated in the forecasting of pen temperatures for animals of other species. Besteiro et al. (2017) developed a model for weaned piglets that employed a complete production cycle as the model estimation stage, resulting in a model that incorporated outdoor ambient temperature as the sole independent variable. In the present study, the THI is employed as a variable, which is not a single independent variable. Rather, it entails the integration of both ambient temperature and relative humidity.

The application of mathematical models for the prediction of factors influencing dairy cattle productivity is becoming increasingly prevalent in the scientific literature. For example, Chavarría et al. (2024) employed ARIMA models to estimate and robustly predict variables such as monthly herd milk production and discard rate. This illustrates the potential of time series modeling of retrospective data to forecast future trends and patterns of development in dairy herds in hot environments.

Pereira et al. (2024) focused on the impact of climate on dairy cattle, evaluating the performance of heat stress classifier models using confusion matrix metrics and contrasting them with the conventional approach based on temperature and humidity indices. The highest accuracy, 86.8%, was achieved, demonstrating the feasibility of developing precise and operational models for real-time monitoring of heat stress. In the context of dairy cattle, Li et al. (2024) employed machine learning techniques through a comprehensive evaluation of multiple feature sets to develop an effective core body temperature prediction model for dairy cows. This approach markedly enhanced the model’s predictive capacity by integrating it with distinctive animal-related attributes and infrared temperature readings, thereby improving its accuracy. These findings may be further considered in other analyses in our region.

The findings suggest the possibility of further research in this region and in other locations experiencing elevated temperatures. The objective is to develop mathematical models that can accurately predict the THI with a high degree of probability. Such knowledge would facilitate the implementation of strategies to mitigate the adverse effects on livestock health and productivity such as scheduling baths at optimal times and switching to lower-calorie diets when elevated THI levels are predicted, may be employed to achieve the desired outcome.

Supplemental Information

Supplemental Information 1 Raw data.

We thank Mr. Jesús S. Rivas Madero, CEO of DiGiTH & DiGiSKY Technologies, for providing the THI values used in this article.

Additional Information and Declarations

Competing Interests

Author Contributions

Data Availability

The authors declare that they have no competing interests.

José Luis Herrera-González conceived and designed the experiments, performed the experiments, authored or reviewed drafts of the article, and approved the final draft.

Rafael Rodríguez-Venegas conceived and designed the experiments, performed the experiments, authored or reviewed drafts of the article, and approved the final draft.

Martín Alfredo Legarreta-González conceived and designed the experiments, performed the experiments, analyzed the data, prepared figures and/or tables, authored or reviewed drafts of the article, and approved the final draft.

Pedro Antonio Robles-Trillo performed the experiments, prepared figures and/or tables, and approved the final draft.

Ángeles De-Santiago-Miramontes performed the experiments, prepared figures and/or tables, and approved the final draft.

Darithsa Loya-González analyzed the data, prepared figures and/or tables, and approved the final draft.

Rafael Rodríguez-Martínez performed the experiments, authored or reviewed drafts of the article, and approved the final draft.

The following information was supplied regarding data availability:

The raw data is available in the Supplemental File.

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
