# Peer review of "Time series (ARIMA) as a tool to predict the temperature-humidity index in the dairy region of the northern desert of Mexico"

_PeerJ, doi:10.7717/peerj.18744_

## Round 0.1 · original submission · Major Revisions

Based on the comments from three anonymous reviewers, your manuscript requires major revision before it can be accepted. Particularly, the relevant knowledge gap, detailed information of the methods, the validation of the prediction model, and more discussion are required to descibe clearly.

Reviewer 1 ·

Basic reporting

The study is of general importance for livestock production, health and welfare. The study examines the predictive capacity of the THI in relation to potential heat stress by using the time series method (ARIMA MODEL). They have predicted THI using four previous days climatic data of previous four years. Accuracy of model prediction for THI has not discussed in detail ?. Results and discussion part is weak in the manuscript. Predictive capacity of model has to be discussed in detail.

Experimental design

Experimental design is okay

Validity of the findings

Validity of the findings is not explained properly.
Furthermore, the data demonstrates an annual increase in the number of days the THI indicates a risk of heat stress. This line is written in abstract, but not discussed in the results and discussion part.

Clearly state what is the gap, hypothesis of study?

Accuracy of model prediction for THI has not discussed in detail ?. Predictive capacity of model with results of model has to be discussed in detail.

Discussion part is not written well. Some points in the Paper has been written very technically, difficult for the readers to comprehend the interpretation.

Generally, it is a very well established fact that Increase in THI can be stressful for the animals. Repeatedly these points in the paper have been written. While the specific ARIMA model utility and its accuracy has not been written at all.

Additional comments

The paper is recommended for major revision.
Many mistakes in reference writing through out the text.

Annotated reviews are not available for download in order to protect the identity of reviewers who chose to remain anonymous.

·

Basic reporting

The article is clear and used academic English.
The literature is especially sufficient in the introduction, but it can be improved in the discussion.
Figures are proper but some explanations should be added (marked in the PDF file). However, it is not clear what the numbers given in the Raw Data file mean.
The hypotheses and the results are relevant.

Experimental design

Experimental design is described with sufficient information. Research question is well defined. However, material and method should be written in a more comprehensive and clear manner that can be repeated in another study. The meaning of each expression given in the formulas should be stated.

Validity of the findings

Findings of this article is important for dairy cattle industry. Because, heat stress is influencing the health, production and welfare of cattle. Therefore, it is significant to predict a-day THI previously to mitigate the adverse effects of heat stress.

Additional comments

Also, some suggestions are given for author in the PDF file.

Reviewer 3 ·

Basic reporting

.

Experimental design

.

Validity of the findings

.

Additional comments

A manuscript submitted for review on the topic: "Time series (ARIMA) as a tool to predict the Temperature-Humidity Index in the dairy region of the northern desert of Mexico" is interesting from a scientific and practical point of view in order to preserve the health and well-being of cows for milk. The authors attempt to predict the occurrence of heat stress in dairy cows.
I have the following notes on the manuscript:
I suggest to the authors, if not already done, that this prediction model be described and filed as a patent proposal before the manuscript is published.
It will be good if the data from the model is compared with the response of animals (milk cows) raised in the region of data collection - body temperature, heart rate, respiration, quantitative and qualitative composition of milk.
It would be good if the manuscript suggested a form in which the model data would be available to farmers so that they could take some cooling measures or some other method to reduce the harmful effect of heat stress on cows before it occurs.
In conclusion: I believe that the manuscript can be accepted for publication in its current form.

---

## Round 0.2 · accepted · Accept

Your manuscript has been accepted for publication in the journal of PeerJ.

Reviewer 1 ·

Basic reporting

To ensure the model's predictions are as precise as possible, the authors have included the information in the paper. It is okay.
Authors have done suggested corrections in the manuscript. Reference writing is also improved except
Sejian,::: V,::::: SMK:::::: Naqvi, :T:::::: Ezeji, :J::::::: Lakritz,::: and::R:::: Lal.::::: 2013.::::::::::::: Environmental::::: Stress:::: and::::::::::: Amelioration:: 429 in
:::::::: Livestock ::::::::: Production.: It is incomplete reference (line no. 429)
Authors have not included the details of publication in citation

Experimental design

It has been improved as per suggestion

Validity of the findings

Done as per suggestion. It is okay.

Additional comments

No comment